# Causal Abstraction Inference under Lossy Representations

Kevin Xia [1]   Elias Bareinboim [1]

## Abstract

The study of causal abstractions bridges two integral components of human intelligence: the ability to determine cause and effect, and the ability to interpret complex patterns into abstract concepts. Formally, causal abstraction frameworks define connections between complicated low-level causal models and simple high-level ones. One major limitation of most existing definitions is that they are not well-defined when considering lossy abstraction functions in which multiple low-level interventions can have different effects while mapping to the same high-level intervention (an assumption called the abstract invariance condition). In this paper, we introduce a new type of abstractions called projected abstractions that generalize existing definitions to accommodate lossy representations. We show how to construct a projected abstraction from the low-level model and how it translates equivalent observational, interventional, and counterfactual causal queries from low to high-level. Given that the true model is rarely available in practice we prove a new graphical criteria for identifying and estimating high-level causal queries from limited low-level data. Finally, we experimentally show the effectiveness of projected abstraction models in high-dimensional image settings.

## 1. Introduction

The ability to determine cause and effect, and the ability to interpret complex patterns into abstract concepts, are two integral components of human intelligence. From the causality perspective, causal reasoning is vital in planning courses of actions, determining blame and responsibility, and generalizing across changing environments. From the abstraction perspective, humans generally grasp better intuition when understanding something at a high-level. For

example, a human can easily parse the object in an image as a dog or a car instead of interpreting it as a collection of pixel values. Combining these two modes of reasoning is vital for building more advanced AI systems.

Causal inference is often studied under the semantics of structural causal models (SCMs) (Pearl, 2000). An SCM models reality with a collection of mechanisms and exogenous distributions. Each SCM induces a collection of distributions categorized into three successively more descriptive layers known as the Ladder of Causation or Pearl Causal Hierarchy (PCH) (Pearl & Mackenzie, 2018; Bareinboim et al., 2022). These three layers refer to the observational ($\mathcal{L}_1$), interventional ($\mathcal{L}_2$), and counterfactual ($\mathcal{L}_3$) distributions. In many causal inference tasks, the goal is to infer a quantity from a higher layer using data from lower layers, a problem known as *cross-layer inference*. It is understood that it is generally impossible to infer higher layer information without additional assumptions (a result known as the Causal Hierarchy Theorem or CHT (Bareinboim et al., 2022)), so understanding the necessary assumptions for performing inferences is a key component of any causal inference task.

Existing works on causal abstractions have made significant progress in defining abstraction principles, proving insightful properties, and learning abstraction functions in practice (Rubenstein et al., 2017; Beckers & Halpern, 2019; Beckers et al., 2019; Geiger et al., 2023; Massidda et al., 2023; Zennaro et al., 2023; Felekis et al., 2024). Causal abstractions are typically studied by comparing a high-level model $\mathcal{M}_H$, defined over high-level variables $\mathbf{V}_H$, with its low-level counterpart $\mathcal{M}_L$, defined over $\mathbf{V}_L$. An abstraction function $\tau$ maps from $\mathbf{V}_L$ to $\mathbf{V}_H$, and $\mathcal{M}_H$ is formally defined as an abstraction of $\mathcal{M}_L$ if it satisfies key properties with respect to $\tau$ such as commutativity with interventions. More recently, this notion has been relaxed to only enforcing properties between distributions of $\mathcal{M}_H$ and $\mathcal{M}_L$ from the PCH (Xia & Bareinboim, 2024). For example, rather than saying $\mathcal{M}_H$ is a full abstraction of $\mathcal{M}_L$, one can say that $\mathcal{M}_H$ is an abstraction of $\mathcal{M}_L$ specifically for interventional quantities in $\mathcal{L}_2$ or for a single causal effect $P(y \mid do(x)) \in \mathcal{L}_2$. Xia & Bareinboim (2024) also shows the synergy between causal abstraction theory and representation learning (Bengio et al., 2013), which has shown great success in many deep learning applications by mapping high-dimensional data like images or text to simpler representation spaces. These definitions of

[1]CausalAI Lab, Columbia University. Correspondence to: Kevin Xia <kmx2000@columbia.edu>.

*Proceedings of the $42^{nd}$ International Conference on Machine Learning*, Vancouver, Canada. PMLR 267, 2025. Copyright 2025 by the author(s).

causal abstractions have accomplished formalizing a broad topic of human intelligence into mathematical language.

One particular limitation of existing definitions of abstractions is known as the Abstract Invariance Condition (AIC), which states, informally, that two values cannot be abstracted together if they have different downstream impacts. This is illustrated in Fig. 1. For example, a nutritionist may have collected data on two types of cholesterol, HDL and LDL, and are studying their impact on heart disease (Steinberg, 2007; Truswell, 2010). They would like to abstract the two together by summing them as total cholesterol (TC). However, this violates the AIC, as it is known that HDL decreases rate of heart disease while LDL increases it, so the sum is ambiguous (a lossy representation).[1] Nonetheless, it may still be desirable to have a consistent formalism in which these kinds of ambiguous abstractions are well-defined, since in many practical settings (where representation learning or dimensionality reduction is needed), the AIC is clearly violated or is impossible to verify.

In this paper, we study this extension of causal abstractions, which we later define as *projected abstractions*, referring to the idea that an abstraction that violates the AIC results in a loss of information that is then characterized in the exogenous space. The proposed formalism generalizes abstractions both on the SCM and on the PCH level to allow for mathematically consistent abstractions even with AIC violations. Projected abstractions have many uses in practice, resulting in tractable causal inference and high-quality causal sampling even in the presence of extreme dimensionality reduction, a result which we show in the experiments.

To summarize, in Sec. 2, we generalize abstractions to settings which the AIC does not hold and provide an algorithm for constructing the high-level model. In Sec. 3, we show how to perform causal inference from data within this class of abstractions when the true model is not observed. In Sec. 4, we empirically demonstrate the power of abstractions at performing causal inference in high-dimensional image settings. All proofs can be found in App. A. Appendices can be found in the full technical report, Xia & Bareinboim (2025).

### 1.1. Preliminaries

We now introduce the notation and definitions used throughout the paper. We use uppercase letters $(X)$ to denote random variables and lowercase letters $(x)$ to denote corresponding values. Similarly, bold uppercase $(\mathbf{X})$ and lowercase $(\mathbf{x})$ letters denote sets of random variables and values respectively. We use $\mathcal{D}_X$ to denote the domain of $X$ and $\mathcal{D}_{\mathbf{X}} = \mathcal{D}_{X_1} \times \cdots \times \mathcal{D}_{X_k}$ for the domain of $\mathbf{X} = \{X_1, \ldots, X_k\}$. We denote $P(\mathbf{X} = \mathbf{x})$ (often short-

---

[1]See App. C Ex. 7 for a more concrete explanation.

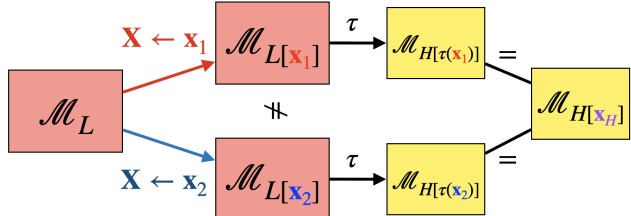

Figure 1: An illustration of AIC violations. On the low level, two different interventions may be performed (e.g., $\mathbf{X} \leftarrow \mathbf{x}_1$ and $\mathbf{X} \leftarrow \mathbf{x}_2$). However, after applying the abstraction function $\tau$ to obtain the high-level model, both interventions are mapped to the same result ($\tau(\mathbf{x}_1) = \tau(\mathbf{x}_2) = \mathbf{x}_H$). If $\mathcal{M}_L$ behaves differently under $\mathbf{x}_1$ compared to $\mathbf{x}_2$, $\mathcal{M}_H$ cannot stay consistent with both models.

ened to $P(\mathbf{x})$) as the probability of $\mathbf{X}$ taking the values $\mathbf{x}$ under the distribution $P(\mathbf{X})$.

We utilize the basic semantic framework of structural causal models (SCMs) (Pearl, 2000), following the presentation in Bareinboim et al. (2022).

**Definition 1** (Structural Causal Model (SCM))**.** An SCM $\mathcal{M}$ is a 4-tuple $\langle \mathbf{U}, \mathbf{V}, \mathcal{F}, P(\mathbf{U}) \rangle$, where $\mathbf{U}$ is a set of exogenous variables (or "latents") that are determined by factors outside the model; $\mathbf{V}$ is a set $\{V_1, V_2, \ldots, V_n\}$ of (endogenous) variables of interest that are determined by other variables in the model – that is, in $\mathbf{U} \cup \mathbf{V}$; $\mathcal{F}$ is a set of functions $\{f_{V_1}, f_{V_2}, \ldots, f_{V_n}\}$ such that each $f_{V_i}$ is a mapping from (the respective domains of) $\mathbf{U}_{V_i} \cup \mathbf{Pa}_{V_i}$ to $V_i$, where $\mathbf{U}_{V_i} \subseteq \mathbf{U}$, $\mathbf{Pa}_{V_i} \subseteq \mathbf{V} \setminus V_i$, and the entire set $\mathcal{F}$ forms a mapping from $\mathbf{U}$ to $\mathbf{V}$. That is, for $i = 1, \ldots, n$, each $f_{V_i} \in \mathcal{F}$ is such that $v_i \leftarrow f_{V_i}(\mathbf{pa}_{V_i}, \mathbf{u}_{V_i})$; and $P(\mathbf{U})$ is a probability function defined over the domain of $\mathbf{U}$. ∎

Each $\mathcal{M}$ induces a causal diagram $\mathcal{G}$, where every $V_i \in \mathbf{V}$ is a vertex, there is a directed arrow $(V_j \to V_i)$ for every $V_i \in \mathbf{V}$ and $V_j \in \mathbf{Pa}_{V_i}$, and there is a dashed-bidirected arrow $(V_j \dashleftarrow\dashrightarrow V_i)$ for every pair $V_i, V_j \in \mathbf{V}$ such that $\mathbf{U}_{V_i}$ and $\mathbf{U}_{V_j}$ are not independent (Markovianity is not assumed). Our treatment is constrained to *recursive* SCMs, which implies acyclic causal diagrams, with finite discrete domains over endogenous variables $\mathbf{V}$.

Counterfactual (and also interventional and observational) quantities can be computed from SCM $\mathcal{M}$ as follows:

**Definition 2** (Layer 3 Valuation (Bareinboim et al., 2022, Def. 7))**.** An SCM $\mathcal{M}$ induces layer $\mathcal{L}_3(\mathcal{M})$, a set of distributions over $\mathbf{V}$, each with the form $P(\mathbf{Y}_*) = P(\mathbf{Y}_{1[\mathbf{x}_1]}, \mathbf{Y}_{2[\mathbf{x}_2], \ldots})$ such that

$$P^{\mathcal{M}}(\mathbf{y}_{1[\mathbf{x}_1]}, \mathbf{y}_{2[\mathbf{x}_2]}, \ldots) = \tag{1}$$

$$\int_{\mathcal{D}_{\mathbf{U}}} \mathbf{1}\left[ \mathbf{Y}_{1[\mathbf{x}_1]}(\mathbf{u}) = \mathbf{y}_1, \mathbf{Y}_{2[\mathbf{x}_2]}(\mathbf{u}) = \mathbf{y}_2, \ldots \right] dP(\mathbf{u})$$

where $\mathbf{Y}_{i[\mathbf{x}_i]}(\mathbf{u})$ is evaluated under $\mathcal{F}_{\mathbf{x}_i} := \{f_{V_j} : V_j \in \mathbf{V} \setminus \mathbf{X}_i\} \cup \{f_X \leftarrow x : X \in \mathbf{X}_i\}$. $\mathcal{L}_2$ is the subset of $\mathcal{L}_3$ for which all $\mathbf{x}_i$ are equal, and $\mathcal{L}_1$ is the subset for which all $\mathbf{X}_i = \emptyset$. ∎

Each $\mathbf{Y}_i$ corresponds to a set of variables in a world where the original mechanisms $f_X$ are replaced with constants $\mathbf{x}_i$ for each $X \in \mathbf{X}_i$; this is also known as the mutilation procedure. This procedure corresponds to interventions, and we use subscripts to denote the intervening variables (e.g. $\mathbf{Y}_\mathbf{x}$) or subscripts with brackets when the variables are indexed (e.g. $\mathbf{Y}_{1[\mathbf{x}_1]}$). For instance, $P(y_x, y'_{x'})$ is the probability of the joint counterfactual event $Y = y$ had $X$ been $x$ and $Y = y'$ had $X$ been $x'$.

We use the notation $\mathcal{L}_i(\mathcal{M})$ to denote the set of $\mathcal{L}_i$ distributions from $\mathcal{M}$. We use $\mathbb{Z}$ to denote a set of quantities from Layer 2 (i.e. $\mathbb{Z} = \{P(\mathbf{V}_{\mathbf{z}_k})\}_{k=1}^\ell$), and $\mathbb{Z}(\mathcal{M})$ denotes those same quantities induced by SCM $\mathcal{M}$ (i.e. $\mathbb{Z}(\mathcal{M}) = \{P^\mathcal{M}(\mathbf{V}_{\mathbf{z}_k})\}_{k=1}^\ell$).

The theory of causal abstractions developed in this paper build on the foundations of constructive abstraction functions, under which individual distributions of the PCH are well-defined between low and high-level models.

**Definition 3** (Inter/Intravariable Clusterings (Xia & Bareinboim, 2024, Def. 5)). Let $\mathcal{M}$ be an SCM over $\mathbf{V}$.

1. A set $\mathbb{C}$ is said to be an intervariable clustering of $\mathbf{V}$ if $\mathbb{C} = \{\mathbf{C}_1, \mathbf{C}_2, \ldots \mathbf{C}_n\}$ is a partition of a subset of $\mathbf{V}$. $\mathbb{C}$ is further considered admissible w.r.t. $\mathcal{M}$ if for any $\mathbf{C}_i \in \mathbb{C}$ and any $V \in \mathbf{C}_i$, no descendent of $V$ outside of $\mathbf{C}_i$ is an ancestor of any variable in $\mathbf{C}_i$. That is, there exists a topological ordering of the clusters of $\mathbb{C}$ relative to the functions of $\mathcal{M}$.

2. A set $\mathbb{D}$ is said to be an intravariable clustering of variables $\mathbf{V}$ w.r.t. $\mathbb{C}$ if $\mathbb{D} = \{\mathbb{D}_{\mathbf{C}_i} : \mathbf{C}_i \in \mathbb{C}\}$, where $\mathbb{D}_{\mathbf{C}_i} = \{\mathcal{D}_{\mathbf{C}_i}^1, \mathcal{D}_{\mathbf{C}_i}^2, \ldots, \mathcal{D}_{\mathbf{C}_i}^{m_i}\}$ is a partition (of size $m_i$) of the domains of the variables in $\mathbf{C}_i$, $\mathcal{D}_{\mathbf{C}_i}$ (recall that $\mathcal{D}_{\mathbf{C}_i}$ is the Cartesian product $\mathcal{D}_{V_1} \times \mathcal{D}_{V_2} \times \cdots \times \mathcal{D}_{V_k}$ for $\mathbf{C}_i = \{V_1, V_2, \ldots, V_k\}$, so elements of $\mathcal{D}_{\mathbf{C}_i}^j$ take the form of tuples of the value settings of $\mathbf{C}_i$). ∎

**Definition 4** (Constructive Abstraction Function (Xia & Bareinboim, 2024, Def. 6)). A function $\tau : \mathcal{D}_{\mathbf{V}_L} \to \mathcal{D}_{\mathbf{V}_H}$ is said to be a constructive abstraction function w.r.t. inter/intravariable clusters $\mathbb{C}$ and $\mathbb{D}$ iff

1. There exists a bijective mapping between $\mathbf{V}_H$ and $\mathbb{C}$ such that each $V_{H,i} \in \mathbf{V}_H$ corresponds to $\mathbf{C}_i \in \mathbb{C}$;

2. For each $V_{H,i} \in \mathbf{V}_H$, there exists a bijective mapping between $\mathcal{D}_{V_{H,i}}$ and $\mathbb{D}_{\mathbf{C}_i}$ such that each $v_{H,i}^j \in \mathcal{D}_{V_{H,i}}$ corresponds to $\mathcal{D}_{\mathbf{C}_i}^j \in \mathbb{D}_{\mathbf{C}_i}$; and

3. $\tau$ is composed of subfunctions $\tau_{\mathbf{C}_i}$ for each $\mathbf{C}_i \in \mathbb{C}$ such that $\mathbf{v}_H = \tau(\mathbf{v}_L) = (\tau_{\mathbf{C}_i}(\mathbf{c}_i) : \mathbf{C}_i \in \mathbb{C})$, where $\tau_{\mathbf{C}_i}(\mathbf{c}_i) = v_{H,i}^j$ if and only if $\mathbf{c}_i \in \mathcal{D}_{\mathbf{C}_i}^j$. We also apply the same notation for any $\mathbf{W}_L \subseteq \mathbf{V}_L$ such that $\mathbf{W}_L$ is a union of clusters in $\mathbb{C}$ (i.e. $\tau(\mathbf{w}_L) = (\tau_{\mathbf{C}_i}(\mathbf{c}_i) : \mathbf{C}_i \in \mathbb{C}, \mathbf{C}_i \subseteq \mathbf{W}_L))$. ∎

Finally, we state the AIC formally below.

**Definition 5** (Abstract Invariance Condition (AIC)). Let $\mathcal{M}_L = \langle \mathbf{U}_L, \mathbf{V}_L, \mathcal{F}_L, P(\mathbf{U}_L) \rangle$ be an SCM and $\tau : \mathcal{D}_{\mathbf{V}_L} \to \mathcal{D}_{\mathbf{V}_H}$ be a constructive abstraction function relative to $\mathbb{C}$ and $\mathbb{D}$. The SCM $\mathcal{M}_L$ is said to satisfy the abstract invariance condition (AIC, for short) with respect to $\tau$ if, for all $\mathbf{v}_1, \mathbf{v}_2 \in \mathcal{D}_{\mathbf{V}_L}$ such that $\tau(\mathbf{v}_1) = \tau(\mathbf{v}_2)$, $\forall \mathbf{u} \in \mathcal{D}_{\mathbf{U}_L}, \mathbf{C}_i \in \mathbb{C}$, the following holds:

$$
\begin{aligned}
&\tau_{\mathbf{C}_i}\left(\left(f_V^L(\mathbf{pa}_V^{(1)}, \mathbf{u}_V) : V \in \mathbf{C}_i\right)\right) \\
&= \tau_{\mathbf{C}_i}\left(\left(f_V^L(\mathbf{pa}_V^{(2)}, \mathbf{u}_V) : V \in \mathbf{C}_i\right)\right),
\end{aligned} \tag{2}
$$

where $\mathbf{pa}_V^{(1)}$ and $\mathbf{pa}_V^{(2)}$ are the values corresponding to $\mathbf{v}_1$ and $\mathbf{v}_2$. ∎

A table summarizing the notation can be found in App. A.1, detailed explanations of these definitions can be found in App. A.2, and additional useful definitions from prior work can be found in App. A.3.

## 2. Abstractions under AIC Violations

The abstract invariance condition (AIC) states, in words, that two low-level values cannot map to the same high-level value if they have different downstream effects. This is a critical property that must hold for existing definitions of abstractions to be well-defined. In this paper, we will use the following running example to illustrate the key points.

**Example 1.** For concreteness, consider a setting in which different insurance companies ($Z$) offer various insurance plans ($X$), which affect whether an insurance claim is approved ($Y$). For simplicity, suppose there are two insurance companies ($z_1$ and $z_2$) that offer three insurance plans ($x_1$, $x_2$, and $x_3$), and the claim is either approved ($Y = 1$) or not approved ($Y = 0$). Suppose the true model $\mathcal{M}^* = \mathcal{M}_L = \langle \mathbf{U}_L, \mathbf{V}_L, \mathcal{F}_L, P(\mathbf{U}_L) \rangle$ is described as

$$
\begin{aligned}
\mathbf{U}_L &= \{U_Z, U_X^{z_1}, U_X^{z_2}, U_Y^{x_1}, U_Y^{x_2}, U_Y^{x_3}\} \\
\mathbf{V}_L &= \{Z, X, Y\} \\
\mathcal{F}_L &= \begin{cases} f_Z^L(u_Z) &= u_Z \\ f_X^L(z, u_X^{z_1}, u_X^{z_2}) &= u_X^z \\ f_Y^L(x, u_Y^{x_1}, u_Y^{x_2}, u_Y^{x_3}) &= u_Y^x \end{cases}
\end{aligned} \tag{3}
$$

$$P(\mathbf{U}_L) = \begin{cases} P(U_Z = z_1) = 0.5 \\ P(U_X^{z_1}) = \{x_1 \to 0.4; x_2 \to 0.1; x_3 \to 0.5\} \\ P(U_X^{z_2}) = \{x_1 \to 0.1; x_2 \to 0.4; x_3 \to 0.5\} \\ P(U_Y^{x_1} = 1) = 0.9, P(U_Y^{x_2} = 1) = 0.1, \\ P(U_Y^{x_3} = 1) = 0.9 \end{cases}$$

The interpretation of the model is as follows: Insurance plans $x_1$ and $x_3$ are very effective, with $0.9$ probability of claim acceptance, while $x_2$ is very ineffective at only $0.1$ probability. Insurance company $z_1$ is more reputable than $z_2$ and is more likely to offer plan $x_1$ over $x_2$, while company $z_2$ prefers to offer plan $x_2$ over $x_1$.

Suppose an important factor of consideration not shown in the model is that $x_1$ and $x_2$ are cheaper insurance plans, while $x_3$ is more expensive. A data scientist who is studying this model may choose to abstract the different plans away, categorizing them simply as "cheap" and "expensive" plans. Formally, they would study a set of higher-level variables $\mathbf{V}_H = \{Z_H, X_H, Y_H\}$, where $Z_H = Z$, $Y_H = Y$, and $X_H$ has a domain $\mathcal{D}_{X_H} = \{x_C, x_E\}$ corresponding to cheap and expensive plans respectively. There exists an abstraction function $\tau : \mathcal{D}_{\mathbf{V}_L} \to \mathcal{D}_{\mathbf{V}_H}$ such that $\tau$ maps $x_1$ and $x_2$ to $x_C$ (cheap) and maps $x_3$ to $x_E$ (expensive). We will sometimes use the notation $X_L$ to describe $X$ to disambiguate from $X_H$, and we will use the notation $Z$ and $Y$ instead of $Z_H$ and $Y_H$ since the variables are the same on both levels.

This immediately brings the AIC into question. If the data scientist is interested in the causal effect of cheap plans on claim acceptance (i.e., $P(Y_{X_H = x_C} = 1)$), whether $x_C$ refers to $x_1$ or $x_2$ is ambiguous. To witness, note that

$$P(Y_{X_L = x_1} = 1) = 0.9 \tag{4}$$
$$P(Y_{X_L = x_2} = 1) = 0.1. \tag{5}$$

Since $\tau(x_1) = \tau(x_2) = x_C$, but $P(Y_{x_1}) \neq P(Y_{x_2})$, the AIC is clearly violated, leaving the intervention on $x_C$ ambiguous. ∎

Fundamentally, the issue with AIC violations is clear: formal definitions of abstractions expect an equality between low-level and corresponding high-level quantities, but it is not well-defined when one high-level quantity corresponds to multiple differing low-level quantities. In practice, the AIC can be a difficult restriction. Generally, it is assumed to be true whenever abstractions are applied, but it is difficult to verify given that the true SCM and functions are rarely available in real-world settings. The assumption is also likely to be incorrect when applying abstractions naïvely, for example, by performing representation learning or dimensionality reduction without taking the AIC into account. By definition, dimensionality reduction is a lossy transformation of the original data, and the AIC is violated if any of the lost information is relevant for downstream functions.

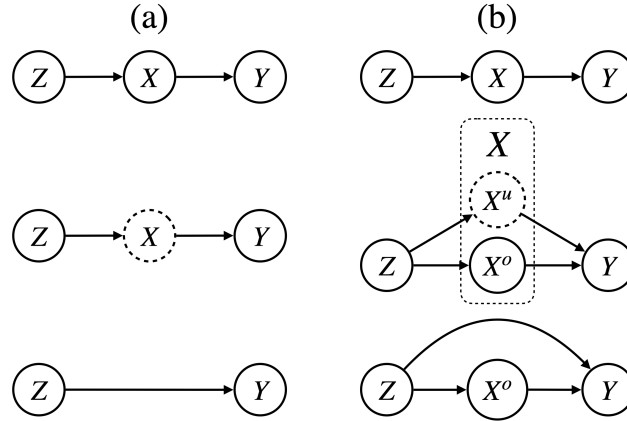

Figure 2: Comparison between (a) full SCM projections and (b) partial SCM projections. When $X$ is fully projected away, its function is subsumed by its child's function $f_Y$. When $X$ is partially projected, it is split into observed portion $X^o$ and unobserved portion $X^u$. The role of $X^o$ is preserved, while $X^u$ is subsumed into the function $f_Y$.

Even when the AIC does not hold, it does not necessarily mean that these lossy transformations should not be used. Representation learning and dimensionality reduction are often performed to improve tractability or interpretability at the cost of some lost information. Hence, it would still be desirable to perform causal inferences in the high-level space even under AIC violations. To address the issue of different low-level quantities matching the same high-level quantity, one can reinterpret the high-level quantity as a distribution over its corresponding low-level quantities, where the randomness in the distribution results from the lost information from the abstraction (i.e., a hard intervention on the high-level translates to a soft intervention on the low-level).

## 2.1. Projected Abstractions

The discussion on relaxing the AIC begins with the concept of SCM projections (Lee & Bareinboim, 2019), which can be viewed as a primitive form of abstraction. An SCM $\mathcal{M}$ projected to a subset of variables $\mathbf{W} \subseteq \mathbf{V}$ is a functionally identical SCM defined over $\mathbf{W}$, where the functions of $\mathbf{V} \setminus \mathbf{W}$ are subsumed by other downstream functions (see App. A Def. 4 for the full definition and App. C Ex. 8 for an example). In the context of constructive abstraction functions, the act of projecting away a variable can be viewed as excluding the variable from all intervariable clusters. This brings the first major insight in addressing AIC violations. In general, when reducing the granularity of a variable, some parts of the variable deemed less important are abstracted away while others are retained. While by definition, SCM projections only allow for entire variables to be included or excluded, one could conceive of SCM projections in which variables are only partially projected away (see App. C Ex. 9

for an example). Formally, partial SCM projections can be defined as follows.

**Proposition 1** (Partial SCM Projection). *Let* $\mathbf{V}$ *be a set of variables and* $\mathbf{W} \subseteq \mathbf{V}$ *be a subset. For each* $W_i \in \mathbf{W}$, *let* $\delta_i : \mathcal{D}_{W_i^o} \times \mathcal{D}_{W_i^u} \to \mathcal{D}_{W_i}$ *be a surjective function mapping new variables* $W_i^o$ *and* $W_i^u$ *to* $W_i$. $W_i^o$ *and* $W_i^u$ *are called the observed and unobserved projections of* $W_i$ *respectively. Denote* $\delta(\mathbf{W}^o, \mathbf{W}^u) = \mathbf{W}$, *where* $\mathbf{W}^o = \{W_i^o : W_i \in \mathbf{W}\}$ *and* $\mathbf{W}^u = \{W_i^u : W_i \in \mathbf{W}\}$. *For any SCM* $\mathcal{M} = \langle \mathbf{U}, \mathbf{V}, \mathcal{F}, P(\mathbf{U}) \rangle$, *there exists an SCM* $\mathcal{M}' = \langle \mathbf{U}' = \mathbf{U} \cup \mathbf{W}^u, \mathbf{V}' = \mathbf{W}^o, \mathcal{F}', P(\mathbf{U}') \rangle$ *such that, for all* $\mathbf{u} \in \mathcal{D}_{\mathbf{U}}$, $\mathbf{X} \subseteq \mathbf{W}$, *and* $\mathbf{x} \in \mathcal{D}_{\mathbf{X}}$,

$$\mathbf{w}_{\mathbf{x}}^o = \mathcal{M}'_{[\mathbf{x}^o]}(\mathbf{u}, \mathbf{x}^u, \mathbf{z}^u), \tag{6}$$

*where* $\delta(\mathbf{w}_{\mathbf{x}}^o, \mathbf{w}_{\mathbf{x}}^u) = \mathbf{W}_{\mathbf{x}}(\mathbf{u})$, $\delta(\mathbf{x}^o, \mathbf{x}^u) = \mathbf{x}$, $\mathbf{Z}^u = \mathbf{W}^u \setminus \mathbf{X}^u$, *and* $\mathbf{z}^u$ *are the corresponding values from* $\mathbf{w}_{\mathbf{x}}^u$. $\mathcal{M}'$ *is called a partial SCM projection of* $\mathcal{M}$ *over* $\mathbf{W}^o$. ∎

In words, a partial SCM projection of $\mathcal{M}$ over $\mathbf{W}^o$ is essentially a smaller version of $\mathcal{M}$ defined only on the variables of $\mathbf{W} \subseteq \mathbf{V}$, where each $W_i \in \mathbf{W}$ is only partially represented in the projection. A function $\delta$ splits $W_i$'s domain into its observed ($W_i^o$) and unobserved ($W_i^u$) portions. Eq. 6 ensures that any value of $\mathbf{W}^o$ obtained from an intervention on the original SCM $\mathcal{M}_{\mathbf{x}}$ will match the corresponding output from $\mathcal{M}'$, when the observed portion of the intervention $\mathbf{x}^o$ is applied to $\mathcal{M}'$, while the unobserved portions of $\mathbf{x}^u$ and $\mathbf{w}^u$ are passed as unobserved arguments to the functions. A comparison between regular SCM projections and partial SCM projections is shown in Fig. 2. The definition of projected abstractions follow.

**Definition 6** (Projected Abstraction). An SCM $\mathcal{M}_H$ is a projected abstraction of $\mathcal{M}_L$ if and only if it is a partial SCM projection of a $\tau$-abstraction (Beckers & Halpern, 2019, Def. 3.13) (also Def. 14 in App. A) of $\mathcal{M}_L$. ∎

To provide intuition for projected abstractions, consider the following example.

**Example 2.** Continuing Example 1, given the setup of Eq. 3, suppose $X_H \in \{x_C, x_E\}$ is given the function

$$f_X^H(z, u_X^{z_1}, u_X^{z_2}) = \begin{cases} x_C & u_X^z \in \{x_1, x_2\} \\ x_E & u_X^z = x_3 \end{cases}, \tag{7}$$

and define $X_H^u \in \{x_1, x_2\}$ as a random variable with distribution

$$P(X_H^u = x_i) = P(X_L = x_i \mid X_L \in \{x_1, x_2\}, z). \tag{8}$$

Suppose now $Y$ is now given a high-level function

$$f_Y^H(x_H, x_H^u, u_Y^{x_i}) = \begin{cases} u_Y^{x_1} & x_H = x_C, X_H^u = x_1 \\ u_Y^{x_2} & x_H = x_C, X_H^u = x_2 \\ u_Y^{x_3} & x_H = x_E \end{cases}. \tag{9}$$

Observe the intuition from constructing these functions from the perspective of projected abstractions. $f_X^H$ behaves identically to $f_X^L$, except the output remaps the value of $X_L$ to the corresponding $X_H$ (i.e. $f_X^H = \tau(f_X^L)$). However, due to the AIC violation, $f_Y^H$ is unable to disambiguate between $x_1$ and $x_2$ if $X_H = x_C$. The solution is to introduce a new exogenous variable $X_H^u$ which represents information in $X_L$ that is not captured in $X_H$ and disambiguates between $x_1$ and $x_2$. $f_Y^H$ then uses both $X_H$ and $X_H^u$ to mimic the behavior of $X_L$. It is clear that $X_L$ can be constructed as $\delta(X_H, X_H^u)$, defined as

$$\delta(x_H, x_H^u) = \begin{cases} x_1 & x_H = x_C, X_H^u = x_1 \\ x_2 & x_H = x_C, X_H^u = x_2 \\ x_3 & x_H = x_E \end{cases}, \tag{10}$$

which matches Eq. 9. Indeed, $\mathcal{M}_H = \langle \mathbf{U}_H = \mathbf{U}_L \cup \{X_H^u\}, \mathbf{V}_H, \mathcal{F}_H = \{f_Z^L, f_X^H, f_Y^H\}, P(\mathbf{U}_H) \rangle$ is a partial SCM projection (and also projected abstraction) of $\mathcal{M}_L$ over $\mathbf{V}_H$. The graph corresponding to $\mathcal{M}_L$ is clearly the top graph of Fig. 2(b), but note that through Eq. 8, there is now a dependence from $Z$ to $Y$, so the graph for $\mathcal{M}_H$ is instead the bottom graph of Fig. 2(b).

It is easy to see that Eq. 6 holds in this example. For instance, fix $U_Z = z_1, U_X^{z_1} = x_2, U_Y^{x_2} = 1$. Clearly, evaluating $\mathcal{M}_L$ with these values results in $Z = z_1, X = x_2, Y = 1$. Note that $x_2 = \delta(x_C, x_2)$, and this is the only set of values of $X_H, X_H^u$ that map to $x_2$. Indeed, on the high level, with $U_Z = z_1, U_X^{z_1} = x_2, U_Y^{x_2} = 1, X_H^u = x_2$, it must also be the case that $Z = z_1, X_H = x_C, Y = 1$. ∎

Projected abstractions make an important step to working around the AIC as Eq. 6 allows for quantities to be well-defined between low and high-level variables by simply obtaining a partial projection of the original SCM $\mathcal{M}_L$ over the high-level variables $\mathbf{V}_H$. However, unlike full SCM projections, partial SCM projections are not unique in terms of the induced PCH distributions. Prop. 1 guarantees its existence but is underspecified in a couple of ways. First, $P(\mathbf{U}')$ is not fully defined, and it is not clear how $\mathbf{W}^u$ should be sampled (e.g., it is not clear how Eq. 8 is chosen in Ex. 2). Second, Eq. 6 does not specify what behavior $\mathcal{M}'$ should follow when $\mathbf{z}^u$ does not match $\mathbf{w}_{\mathbf{x}}^u$ (e.g., How should $Y$ depend on $X_H^u$ in Ex. 2 if $X_H = x_E$?).

The specific choice of partial SCM projection that best serves as an abstraction can be determined by understanding how low-level interventions relate to high-level interventions. In other words, given a high-level intervention $\mathbf{X}_H \leftarrow \mathbf{x}_H$, it is important to define the corresponding low-level soft-intervention $\sigma_{\mathbf{X}_L}$, which is a distribution over all possible interventions $\mathbf{x}_L$ that map to $\mathbf{x}_H$. The consequence of the underspecification of partial SCM projections is that there are many possible choices of defining $\sigma_{\mathbf{X}_L}$. For a full

discussion on how $\sigma_{\mathbf{X}_L}$ should be decided, see App. B. A useful general form of $\sigma_{\mathbf{X}_L}$ is defined as follows. Split $\sigma_{\mathbf{X}_L}$ into individual soft interventions $\sigma_{\mathbf{C}_i}$ for each intervariable cluster $\mathbf{C}_i \subseteq \mathbf{X}_L$. Then define each $\sigma_{\mathbf{C}_i}$ as

$$P(\sigma_{\mathbf{C}_i} = \mathbf{c}_i) = P(\mathbf{c}_i \mid \tau(\mathbf{c}_i) = v_{H,i}, \mathbf{pa}_{V_{H,i}}, \mathbf{u}^c_{V_{H,i}}). \tag{11}$$

In words, a high-level intervention should be equivalent to a distribution over the corresponding low-level interventions that assigns probability to each possible intervention based on their prior probabilities given their parents.[2]

**Example 3.** Continuing Example 1, suppose the data scientist is interested in the causal effect of choosing a cheap insurance plan on claim approval. In other words, she would like to study the intervention $X_H \leftarrow x_C$, which is ambiguous on the low-level as it could refer to either $X_L \leftarrow x_1$ or $X_L \leftarrow x_2$. More specifically, according to Eq. 11, $X_H \leftarrow x_C$ corresponds to a soft intervention $\sigma_{X_C}$ on the low level, defined as

$$\sigma_{X_L} = \begin{cases} x_1 & \text{w.p. } P(x_1 \mid X_L \in \{x_1, x_2\}, z) \\ x_2 & \text{w.p. } P(x_2 \mid X_L \in \{x_1, x_2\}, z) \end{cases} \tag{12}$$

While there are many ways to disambiguate whether $x_C$ is referring to $x_1$ or $x_2$, this choice of $\sigma_{X_L}$ will assign probabilities based on the prior probabilities of $X_L$ being one of $x_1$ or $x_2$. Moreover, the probabilities change depending on the value of $z$. This makes intuitive sense, since under the intervention $X_H \leftarrow x_C$, we expect that if $Z = z_1$, then $X_L$ is more likely to be $x_1$ than $x_2$, or vice-versa when $Z = z_2$. From a query perspective, this implies that

$$P(Y_{X_H = x_C} = 1 \mid Z = z_1) \tag{13}$$
$$= P(Y_{\sigma_{X_L}(x_C, Z)} = 1 \mid Z = z_1)$$
$$= \sum_{x_i \in \{x_1, x_2\}} P(x_i \mid X_L \in \{x_1, x_2\}, z_1) P(Y_{x_i} = 1) = 0.74$$

Likewise, $P(Y_{X_H = x_C} = 1 \mid Z = z_2) = 0.26$ $\tag{14}$ ∎

While projected abstractions are defined over the entire SCM, the mapping between low and high-level interventions are more clear at the query-level (i.e., individual interventional and counterfactual distributions of interest). Such quantities can be defined as follows.

**Definition 7** (Generalized Query). Denote $\mathbf{Y}_{L,*}$ as a set of counterfactual variables over $\mathbf{V}_L$. That is,

$$\mathbf{Y}_{L,*} = \left( \mathbf{Y}_{L,1[\sigma_{\mathbf{X}_{L,1}}]}, \mathbf{Y}_{L,2[\sigma_{\mathbf{X}_{L,2}}]}, \dots \right), \tag{15}$$

---

[2]Here, $\mathbf{u}^c_{V_{H,i}}$ can informally be thought of as the confounded exogenous parents of $V_{H,i}$. The full definition is somewhat involved, and the subtleties are discussed in App. B.2. Due to space constraints, the main body provides intuition in Markovian settings, where unobserved confounding is not present.

**Algorithm 1** Constructing $\mathcal{M}_H$ from $\mathcal{M}_L$.

---
**input** $\mathcal{M}_L = \langle \mathbf{U}_L, \mathbf{V}_L, \mathcal{F}_L, P(\mathbf{U}_L) \rangle$, constructive abstraction function $\tau$ from clusters $\mathbb{C}$ and $\mathbb{D}$
1: $\mathbf{U}_H \leftarrow \mathbf{U}_L, P(\mathbf{U}_H) \leftarrow P(\mathbf{U}_L)$
2: $\mathbf{V}_H \leftarrow \mathbb{C}, \mathcal{D}_{\mathbf{V}_H} \leftarrow \mathbb{D}$
3: **for** $W \in \mathbf{V}_L$ **do**
4: $\quad W^o, W^u \leftarrow \texttt{project}(W)$ {construct $\delta$ from Prop. 1}
5: $\quad \mathbf{U}_H \leftarrow \mathbf{U}_H \cup \{W^u\}$
6: **end for**
7: **for** $\mathbf{C}_i \in \mathbb{C}$ (and corresponding $V_i \in \mathbf{V}_H$) **do**
8: $\quad P(\delta(\mathbf{c}_i^o, \mathbf{C}_i^u) = \mathbf{c}_i \mid \mathbf{U}_L) \leftarrow P(\mathbf{C}_i = \mathbf{c}_i \mid \tau(\mathbf{c}_i) = v_i, \mathbf{pa}_{V_i}, \mathbf{u}^c_{V_{H,i}})$ {from Eq. 11}
9: $\quad f_i^H \leftarrow \tau(f_V^L(\delta(\mathbf{pa}_V^o, \mathbf{pa}_V^u), \mathbf{u}_V) : V \in \mathbf{C}_i)$
10: **end for**
11: $\mathcal{F}_H \leftarrow \{f_i^H : \mathbf{C}_i \in \mathbb{C}\}$
12: **return** $\mathcal{M}_H = \langle \mathbf{U}_H, \mathbf{V}_H, \mathcal{F}_H, P(\mathbf{U}_H) \rangle$

---

where each $\mathbf{Y}_{L,i[\sigma_{\mathbf{X}_{L,i}}]}$ corresponds to the potential outcomes of the variables $\mathbf{Y}_{L,i}$ under the (possibly soft) intervention $\sigma_{\mathbf{X}_{L,i}}$ over $\mathbf{X}_{L,i}$. Each $\mathbf{Y}_{L,i}$ and $\mathbf{X}_{L,i}$ must be unions of clusters from $\mathbb{C}$ (i.e. $\mathbf{Y}_{L,i} = \bigcup_{\mathbf{C} \in \mathbb{C}'} \mathbf{C}$ for some $\mathbb{C}' \subseteq \mathbb{C}$ such that $\tau(\mathbf{Y}_{L,i})$ and $\tau(\mathbf{X}_{L,i})$ are well-defined (i.e. $\tau(\mathbf{Y}_{L,i}) = (\bigwedge_{\mathbf{C} \in \mathbb{C}'} \tau_{\mathbf{C}}(\mathbf{C}))$). For the high-level counterpart, denote

$$\mathbf{Y}_{H,*} = \tau(\mathbf{Y}_{L,*}) = \left( \mathbf{Y}_{H,1[\mathbf{x}_{H,1}]}, \mathbf{Y}_{H,2[\mathbf{x}_{H,2}]}, \dots \right), \tag{16}$$

such that $\mathbf{Y}_{H,i} = \tau(\mathbf{Y}_{L,i})$, and $\mathbf{X}_{H,i} = \tau(\mathbf{X}_{L,i})$ for all $i$. For any value $\mathbf{y}_{H,*} \in \mathcal{D}_{\mathbf{Y}_{H,*}}$, denote

$$\mathcal{D}_{\mathbf{Y}_{L,*}}(\mathbf{y}_{H,*}) = \{\mathbf{y}_{L,*} : \mathbf{y}_{L,*} \in \mathcal{D}_{\mathbf{Y}_{L,*}}, \tau(\mathbf{y}_{L,*}) = \mathbf{y}_{H,*}\}, \tag{17}$$

that is, the set of all values $\mathbf{y}_{L,*}$ such that $\tau(\mathbf{y}_{L,*}) = \mathbf{y}_{H,*}$. For any high-level query

$$\tau(Q) = P(\mathbf{Y}_{H,*} = \mathbf{y}_{H,*}), \tag{18}$$

of the form of Eq. 16, its low-level counterpart is

$$Q = \sum_{\mathbf{y}_{L,*} \in \mathcal{D}_{\mathbf{Y}_{L,*}}(\mathbf{y}_{H,*})} P(\mathbf{Y}_{L,*} = \mathbf{y}_{L,*}), \tag{19}$$

of the form of Eq. 15. ∎

This query definition connects the distributions of $\mathcal{L}_3(\mathcal{M}_H)$ to corresponding distributions of $\mathcal{L}_3(\mathcal{M}_L)$. Compared to earlier definitions, Eq. 15 has been generalized to account for soft interventions in addition to hard interventions. Under constructive abstractions functions $\tau$, a notion of $Q$-$\tau$ consistency was established for certain queries $Q \in \mathcal{L}_3(\mathcal{M}_L)$ (App. A Def. 17), which still apply under this generalized definition. In short, given a low level query $Q$ (Eq. 19) and its high-level counterpart $\tau(Q)$ (Eq. 18), $\mathcal{M}_H$ is said to be $Q$-$\tau$ consistent with $\mathcal{M}_L$ if $Q^{\mathcal{M}_L} = \tau(Q)^{\mathcal{M}_H}$. One can then say that $\mathcal{M}_H$ is an abstraction of $\mathcal{M}_L$ specifically for the query $Q$, even if $\mathcal{M}_H$ may not be $Q'$-$\tau$ consistent with $\mathcal{M}_L$ for other query choices $Q'$. If $\mathcal{M}_H$ is $Q$-$\tau$

consistent with $\mathcal{M}_L$ for all $\tau(Q) \in \mathcal{L}_i(\mathcal{M}_H)$, then $\mathcal{M}_H$ is said to be $\mathcal{L}_i$-$\tau$ consistent with $\mathcal{M}_L$.

With $\sigma_{\mathbf{X}_{L,i}}$ defined in Eq. 11, one can then algorithmically construct a projected abstraction consistent in all queries. Given $\mathcal{M}_L$ and a constructive abstraction function $\tau$ (which may not satisfy the AIC), Alg. 1 can be used to construct the high-level abstraction $\mathcal{M}_H$. In line 4, each $W \in \mathbf{V}_L$ is split into its observed and unobserved counterparts $W^o$ and $W^u$. Line 8 assigns each $W^u$ a distribution based on Eq. 11. Line 9 builds the high-level function using the low-level function with inputs reconstructed using $\delta$. Finally, the full high-level model $\mathcal{M}_H$ is assembled and returned in line 10. Under these inputs, Alg. 1 constructs a projected abstraction $\mathcal{M}_H$ that is $Q$-$\tau$ consistent with $\mathcal{M}_L$ for all possible high-level $\mathcal{L}_3$ queries, as shown by the following result.

**Theorem 1.** *The SCM $\mathcal{M}_H$ constructed by Alg. 1 is a projected abstraction of $\mathcal{M}_L$ that is $Q$-$\tau$ consistent with $\mathcal{M}_L$ for all $\tau(Q) \in \mathcal{L}_3(\mathcal{M}_H)$.* ∎

As an example, it can be verified that running Alg. 1 on $\mathcal{M}_L$ in Ex. 1 results in the SCM $\mathcal{M}_H$ from Ex. 2.

# 3. Projected Abstraction Inference

Alg. 1 finds an abstraction model $\mathcal{M}_H$ that is consistent with its low-level counterpart $\mathcal{M}_L$ for all queries, but it requires the full specification of $\mathcal{M}_L$. In practice, $\mathcal{M}_L$ typically represents the true model of reality and will not be observed. Inferences of $\mathcal{L}_2$ and $\mathcal{L}_3$ queries must be made through limited available data, usually observational ($\mathcal{L}_1$).

The Causal Hierarchy Theorem (Bareinboim et al., 2022, Thm. 1) states that cross-layer inference, or inferring higher layer quantities (e.g., $\mathcal{L}_2$, $\mathcal{L}_3$) from lower layer data (e.g., $\mathcal{L}_1$), is generally impossible without additional assumptions. Many such assumptions take the form of a graphical model, such as a causal diagram (Pearl, 1995), which imply constraints between causal distributions from causal (Bareinboim et al., 2022) and counterfactual Bayesian networks (Correa & Bareinboim, 2024). In the context of abstractions, when $\tau$ is a constructive abstraction function that satisfies the AIC, it has been shown that one can avoid assuming the entire causal diagram of the low-level model in favor of a cluster causal diagram (C-DAG) (Anand et al., 2023) w.r.t. the intervariable clusters $\mathbb{C}$. Unfortunately, this graphical model is insufficient for the case when the AIC is violated.

**Proposition 2** (C-DAG Insufficiency (Informal)). *For a constructive abstraction function $\tau$ over intervariable clusters $\mathbb{C}$ in which the AIC does not hold, the C-DAG $\mathcal{G}_{\mathbb{C}}$ implies constraints that may be unsound.* ∎

To witness why this is the case, Fig. 2(b) shows the issue clearly. Attempting an abstraction in violation of the AIC is akin to performing a partial SCM projection, which may

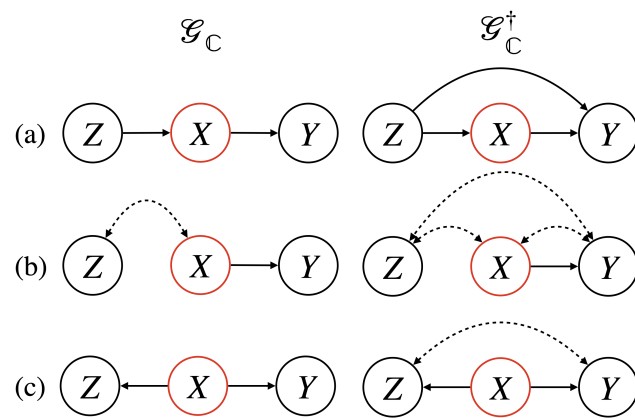

Figure 3: Examples of C-DAGs (left) and their corresponding projected C-DAGs (right), with AIC violation variables $\mathbf{V}_H^\dagger$ outlined in red.

introduce new dependencies between SCM functions, therefore implying new edges in the graph. Ex. 3 explains this dependence numerically. Since no variables are clustered together in the example, both the original causal diagram $\mathcal{G}$ and the C-DAG $\mathcal{G}_{\mathbb{C}}$ are represented by the top graph in Fig. 3. However, this graph implies that $P(Y_{x_H} \mid z) = P(Y_{x_H})$. Evidently, this is not true since Eq. 13 is not equal to Eq. 14. As hinted by the construction in Alg. 1, the high-level function $f_Y^H$ requires some additional information from $Z$ to decide between interpreting $x_C$ as $x_1$ or $x_2$. This information adds a dependence from $Z$ to the function of $f_Y^H$, which requires adding a directed edge from $Z$ to $Y$.

While the original C-DAG construction is not valid for projected abstraction inferences, one can use a modified version that adds the new required dependencies into the C-DAG.

**Definition 8** (Partially Projected C-DAG). Let $\tau : \mathcal{D}_{V_L} \to \mathcal{D}_{V_H}$ be a constructive abstraction function w.r.t. intervariable clusters $\mathbb{C}$ and intravariable clusters $\mathbb{D}$. Let $\mathcal{G}_{\mathbb{C}} = \langle \mathbf{V}_H, \mathbf{E}_{\mathbb{C}} \rangle$ be a C-DAG (with nodes $\mathbf{V}_H$ and edges $\mathbf{E}_{\mathbb{C}}$), of graph $\mathcal{G}$ w.r.t. $\mathbb{C}$. Let $\mathbf{V}_H^\dagger \subseteq \mathbf{V}_H$ be the set of AIC violation variables (App. A Def. 19). Then, construct $\mathcal{G}_{\mathbb{C}}^\dagger = \langle \mathbf{V}_H, \mathbf{E}_{\mathbb{C}}^\dagger \rangle$ as follows. Start by setting $\mathbf{E}_{\mathbb{C}}^\dagger \leftarrow \mathbf{E}_{\mathbb{C}}$. Then apply the following rules for all $X \in \mathbf{V}_H^\dagger$.
(1) If $Z \to X \to Y$ in $\mathbf{E}_{\mathbb{C}}$, then add $Z \to Y$ into $\mathbf{E}_{\mathbb{C}}^\dagger$.
(2) If $Z \leftarrow\!\!\dashrightarrow X \to Y$ in $\mathbf{E}_{\mathbb{C}}$, then add $Z \leftarrow\!\!\dashrightarrow Y$ and $X \leftarrow\!\!\dashrightarrow Y$ into $\mathbf{E}_{\mathbb{C}}^\dagger$.
(3) If $Z \leftarrow X \to Y$ in $\mathbf{E}_{\mathbb{C}}$, then add $Z \leftarrow\!\!\dashrightarrow Y$ into $\mathbf{E}_{\mathbb{C}}^\dagger$.
Repeat iteratively to accommodate new edges.[3] $\mathcal{G}_{\mathbb{C}}^\dagger$ is called the partially projected C-DAG of $\mathcal{G}$ w.r.t. $\mathbb{C}$ and $\mathbf{V}_H^\dagger$. ∎

The steps correspond to the intuition discussed earlier–when performing a partial projection, parts of the variables in $\mathbf{V}_H^\dagger$

---

[3]Procedure can be applied algorithmically in one pass by applying all rules for each node in $\mathbf{V}_H^\dagger$ in topological order.

are projected into the exogenous space, resulting in additional dependences that require additional edge connections. Examples of C-DAGs and their corresponding projected C-DAGs are shown in Fig. 3. In the figure, rows (a), (b), and (c) correspond to examples of steps 1, 2, 3 respectively. It turns out that this new definition is precisely what is needed for abstraction inference in the absence of the AIC.

**Theorem 2** (Projected C-DAG Sufficiency and Necessity (Informal)). *Let $\mathcal{M}_L$ be an SCM over variables $\mathbf{V}_L$, $\tau : \mathcal{D}_{\mathbf{V}_L} \rightarrow \mathcal{D}_{\mathbf{V}_H}$ be a constructive abstraction function w.r.t. clusters $\mathbb{C}$ and $\mathbb{D}$, and $\mathbf{V}_H^{\dagger}$ be the AIC violation set. The partially projected C-DAG $\mathcal{G}_{\mathbb{C}}^{\dagger}$ w.r.t. $\mathbb{C}$ and $\mathbf{V}_H^{\dagger}$ completely describes all constraints over $\mathbf{V}_H$.* ∎

In other words, the projected C-DAG provides exactly the constraints necessary to solve the task of performing causal inferences across abstractions, even when the AIC is violated. In particular, certain interventional and counterfactual distributions may be inferrable from a combination of the projected C-DAG $\mathcal{G}_{\mathbb{C}}^{\dagger}$ and the available datasets from $\mathcal{M}_L$. Determining precisely which queries can be inferred is known as the identification problem, which is defined below in the context of abstract identification.

**Definition 9** (Abstract Identification (General)). Let $\tau : \mathcal{D}_{\mathbf{V}_H} \rightarrow \mathcal{D}_{\mathbf{V}_L}$ be a constructive abstraction function. Consider projected C-DAG $\mathcal{G}_{\mathbb{C}}^{\dagger}$, and let $\mathbb{Z} = \{P(\mathbf{V}_{L[\mathbf{z}_k]})\}_{k=1}^{\ell}$ be a collection of available interventional (or observational if $\mathbf{Z}_k = \emptyset$) distributions over $\mathbf{V}_L$. Let $\Omega_L$ and $\Omega_H$ be the space of SCMs defined over $\mathbf{V}_L$ and $\mathbf{V}_H$, respectively, and let $\Omega_L(\mathcal{G}_{\mathbb{C}}^{\dagger})$ and $\Omega_H(\mathcal{G}_{\mathbb{C}}^{\dagger})$ be their corresponding subsets that induce $\mathcal{G}_{\mathbb{C}}^{\dagger}$. A query $Q$ is said to be $\tau$-ID from $\mathcal{G}_{\mathbb{C}}^{\dagger}$ and $\mathbb{Z}$ iff for every $\mathcal{M}_L \in \Omega_L(\mathcal{G}_{\mathbb{C}}^{\dagger})$, $\mathcal{M}_H \in \Omega_H(\mathcal{G}_{\mathbb{C}}^{\dagger})$ such that $\mathcal{M}_H$ is $\mathbb{Z}$-$\tau$ consistent with $\mathcal{M}_L$, $\mathcal{M}_H$ is also $Q$-$\tau$ consistent with $\mathcal{M}_L$. ∎

In words, a query $Q$ is considered $\tau$-ID if, for any pair of models $\mathcal{M}_L$ and $\mathcal{M}_H$ such that both are compatible with $\mathcal{G}_{\mathbb{C}}^{\dagger}$ and $\mathbb{Z}$, they also match in $Q$. In contrast, $Q$ is not $\tau$-ID if there exist $\mathcal{M}_L$ and $\mathcal{M}_H$ that are compatible with both $\mathcal{G}_{\mathbb{C}}^{\dagger}$ and $\mathbb{Z}$ but disagree on $Q$ (i.e., $Q^{\mathcal{M}_L} \neq \tau(Q)^{\mathcal{M}_H}$). Abstract identification may seem like a difficult property to check, but it turns out that there is a natural connection with the classical identification problem, as shown below.

**Theorem 3** (Dual Abstract ID (General)). *Consider a counterfactual query $Q$ over $\mathbf{V}_L$, a constructive abstraction function $\tau$ w.r.t. clusters $\mathbb{C}$ and $\mathbb{D}$, a projected C-DAG $\mathcal{G}_{\mathbb{C}}^{\dagger}$, and data $\mathbb{Z}$ from $\mathbf{V}_L$. $Q$ is $\tau$-ID from $\mathcal{G}_{\mathbb{C}}^{\dagger}$ and $\mathbb{Z}$ if and only if $\tau(Q)$ is ID from $\mathcal{G}_{\mathbb{C}}^{\dagger}$ and $\tau(\mathbb{Z})$.* ∎

In words, $\tau$-identification across abstractions is equivalent to classic identification on the high-level space.

**Example 4.** Continuing Ex. 1, note that $X_H$ is the only AIC violator in $\mathbf{V}_H$, since $x_1$ and $x_2$ both map to $x_C$ but

have different effects on $Y$. Hence, $\mathbf{V}_H^{\dagger} = \{X_H\}$, and the C-DAG $\mathcal{G}_{\mathbb{C}}$ and projected C-DAG $\mathcal{G}_{\mathbb{C}}^{\dagger}$ are the two graphs in Fig. 3(a). To answer the query of interest $P(Y_{X_H = x_C} = 1)$, one can apply Thm. 3 to simply identify the quantity w.r.t. $P(\mathbf{V}_H)$ and $\mathcal{G}_{\mathbb{C}}^{\dagger}$. In this case, note that the causal effect of $X_H$ on $Y$ can be computed via backdoor adjustment on $Z$, so $P(Y_{X_H = x_C} = 1)$ is equal to

$$\sum_z P(Y = 1 \mid X_H = x_C, Z = z)P(Z = z) \quad (20)$$

$$= \sum_z P(Y = 1 \mid X_L \in \{x_1, x_2\}, z)P(z) \quad (21)$$

$$= (0.7)(0.74) + (0.3)(0.26) = 0.596. \quad (22)$$

∎

Thm. 3 implies that, in practice, $\tau$-ID can be checked by performing any classical ID procedure on the high-level space. This may include algorithmic approaches or other optimization-based approaches.

## 4. Experiments

We perform two experiments to demonstrate the benefits of projected abstractions. The models in the experiments leverage Neural Causal Models (NCMs) (Xia et al., 2021; 2023), specifically the generative adversarial implementation called GAN-NCMs. Details of the experiment setup can be found in App. D, and code can be found at https://github.com/CausalAILab/ProjectedCausalAbstractions.

In the first experiment, we test the necessity of the projected C-DAGs when the AIC does not hold. The high-level query $\tau(Q) = P(y_x \mid z)$ is estimated in the graph setting shown in Fig. 3(a), where $Z$ is a digit from 0 to 9, $X$ is a corresponding colored MNIST image, and $Y$ is a label denoting the color prediction of $X$. $\tau(X)$ maps the image to a binary variable representing the shade (light or dark) of $X$.

The results are shown in Fig. 6. Three different GAN-NCMs are trained: one directly on the low-level data that does not use abstractions (red), an abstracted one constrained by the C-DAG (yellow), and an abstracted one constrained by the projected C-DAG (blue). 95% confidence intervals of the errors are plotted in the figure. Note that the abstractionless model and the projected C-DAG model have decreasing error with more samples, but the regular C-DAG model is unable to learn the correct query. The abstractionless model has higher error than the projected C-DAG model since it operates in a higher-dimensional space.

In the second experiment, we test an interesting consequence of the projected abstraction theory: the soft intervention definition in Eq. 11 can be directly modeled and sampled if attempting to reconstruct the low-level data. We call this

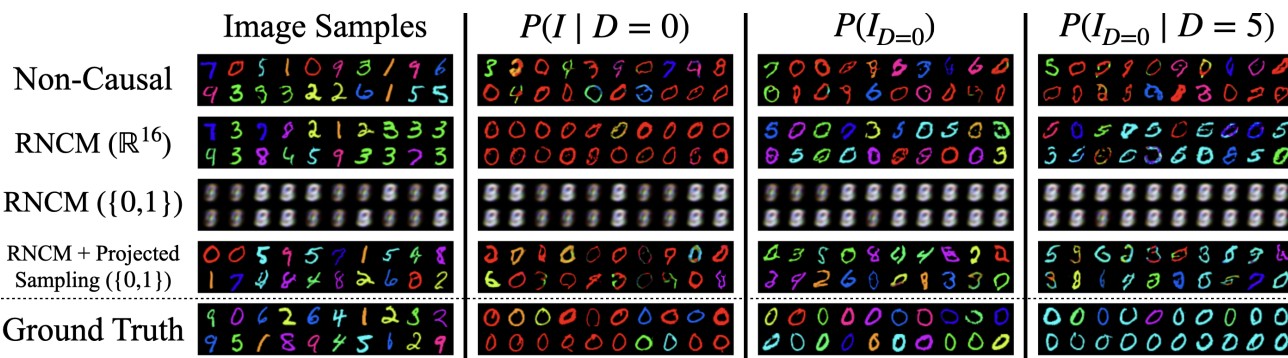

Figure 4: Colored MNIST results. Samples from different causal queries (top) are collected from competing approaches (left). The expressions in parentheses are the representation sizes. The left column shows direct image samples from each of the models, while the second, third, and fourth columns show samples generated from an $\mathcal{L}_1$, $\mathcal{L}_2$, and $\mathcal{L}_3$ query, respectively.

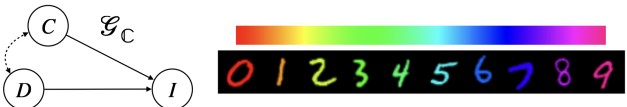

Figure 5: (Left) Graph of Colored MNIST experiment. (Right) Correlation shown between color and digit.

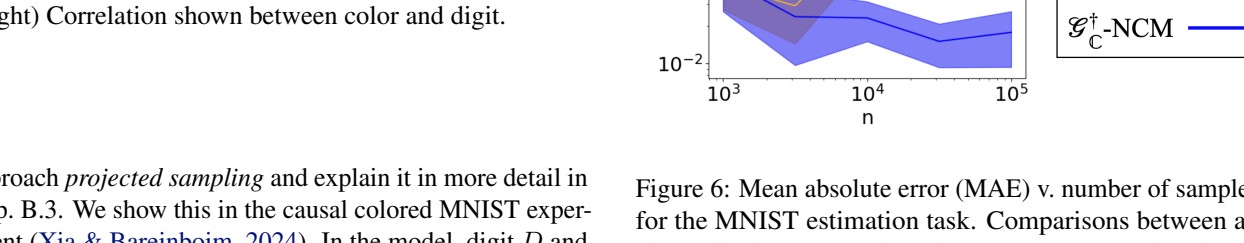

Figure 6: Mean absolute error (MAE) v. number of samples for the MNIST estimation task. Comparisons between an abstractionless approach (red), a C-DAG approach (yellow), and a projected C-DAG approach (blue).

approach *projected sampling* and explain it in more detail in App. B.3. We show this in the causal colored MNIST experiment (Xia & Bareinboim, 2024). In the model, digit $D$ and color $C$ both cause the image $I$, but they are confounded (e.g., 0's are red, 5's are cyan, see Fig. 5). Three different queries are tested (the right three columns of Fig. 4). $P(I \mid D = 0)$ is an $\mathcal{L}_1$ query representing images conditioned on digit = 0, resulting in red 0's. $P(I_{D=0})$ is an $\mathcal{L}_2$ query representing images with the digit intervened as 0, cutting the confounding and resulting in 0's of all colors. $P(I_{D=0} \mid D = 5)$ is an $\mathcal{L}_3$ query representing images with digit intervened as 0, conditioned on the digit originally being 5. This results in 0's with colors of images that were originally 5's, resulting in cyan 0's.

Four methods are compared on these queries in Fig. 4, with the ground truth shown on row 5. The non-causal approach (row 1) simply directly models the conditional distribution between digit and image and therefore fails to model anything higher than $\mathcal{L}_1$. The representational NCM or RNCM (Xia & Bareinboim, 2024) (row 2) is able to decently reproduce all queries, but it uses a 16-dimensional representation space, which cannot shrink much further due to AIC limitations. When forced to take a binary representation (row 3), the RNCM clearly lacks the representation power to properly generate images. In contrast, using a projected sampling approach (row 4) can reproduce the images even with a representation size as small as a binary digit.

## 5. Conclusion

This paper introduced projected abstractions (Def. 6), which can be constructed algorithmically (Alg. 1, Thm. 1), to overcome the AIC limitation. When the full model was not available, we leveraged a new graphical model (Def. 8, Thm. 2) that allowed for causal inferences through the abstract-ID problem (Def. 9, Thm. 3). Finally, we demonstrated the ability of projected abstractions to leverage representation learning within difficult causal inference settings through high-dimensional image experiments.

## Impact Statement

This paper presents work whose goal is to advance the field of causal inference, a subfield of machine learning. The results in this paper may have implications bringing together strong practical results in representation learning and computer vision research with the explainability and generalizability of causal inference results. The trend is that this will lead to smarter AI, which itself has many consequences out

of the scope of this work, but the benefit of understanding causal inference is that it can lead to less bias and more accountability of AI models.

## Acknowledgements

This research is supported in part by the NSF, ONR, AFOSR, DoE, Amazon, JP Morgan, and The Alfred P. Sloan Foundation.

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
