# OpenReview forum: "Causal Abstraction Inference under Lossy Representations"
_ICML.cc/2025/Conference — ICML 2025 poster_

### Official Review · Reviewer_jmSk · 2025-03-12

**Overall Recommendation:** 1

**Summary:**

This paper introduces projected abstractions and an algorithm to compute them from a low-level SCM.

### Update after rebuttal
First of all apologies for an initial review that may have read harsh.

So I carefully read the other reviewers' opinions and the rebuttals. It didn't really bring much more clarity. My impression is that there is even some divergence in what the reviewers think the contribution of the paper is. In parts it seems that even the formalization is considered as part of the contribution, where I think that notation-wise most is taken from what "Neural Causal Abstractions", which I believe is the authors' previous work.

I really do not know what to do with this work. I am convinced that the general research direction of the paper is important, and maybe so is even the contribution. But, at least in the way how it is written down, I can hardly imagine that the paper will have a huge impact beyond a very limited audience that will really make the effort to understand the work in detail. I think that the paper might even be relevant not only for theoretical scientists, so the potential contribution is beyond mere theory I would say. I think it is interesting for me and wouldn't consider myself a theoretic researcher in this field.

So my concerns about clarity prevail and I am going to leave my assessment as is, but apparently I am the outlier. If for the other reviewers the paper not only makes a relevant technical contribution but also manages to properly convey it in their opinion, I have no trouble with the paper being accepted.

**Claims And Evidence:**

I am unsure what significant claims this paper wants to make. There is the introduction of the computation of projected abstractions, which is probably a straight forward algorithm once one buys the ton of definitions of projections. There is also the claim that those projected abstractions "make an important step around the AIC", but it is not entirely clear to me neither what this is supposed to mean exactly and how the algorithm assures it. I am not saying that the experiments don't deliver on a certain expectation, I am just saying that the paper fails to make the contribution understandable, even to someone with fairly solid knowledge in causal inference.

**Essential References Not Discussed:**

None I am aware of.

**Experimental Designs Or Analyses:**

n/a

**Methods And Evaluation Criteria:**

n/a

**Other Comments Or Suggestions:**

Let someone familiar with causal inference (and maybe abstractions) read your paper before you submit to ICML and check on clarity.

I beg the authors to try to make things simpler notation-wise or be at least more didactic and structured in the write-up. I understand that there are two layers of abstraction going on, and defining mappings between them considering all relevant details can be tough. But my experience tells me that probably some definitions could be merged and even some symbols ommitted.

It is urgently necessary that language is used more carefully. In the motivation (line 70-72 left), the authors say "it may be desirable to have a formalism in which these kinds of ambiguous abstractions are well-defined". I strongly doubt that the word "well-defined" is the correct choice here. My current intuition is that with a different type of abstraction you have other *properties* that allow you to do something that you cannot do with previous approaches. If your contribution is not more than making something well-defined, then we seriously have an issue.

In this line, the contribution needs to become clearer. And with this I do not mean to make an explicit bullet list of 3 un-understandable points, but to very concretely formulate a limitation of previous approaches and how you overcome it. I believe that example 1 could help a lot in this regard if one introduces it on an intuitive (already somewhat formal) level without the need of having the AIC conditioned formalized already at that point. It is really weird that we have a 2.5 page introduction and the relevant example only follows in section 2.

**Other Strengths And Weaknesses:**

The presentation of the paper is in my view not top-tier conference-ready. In general the paper is very difficult to read, even for someone who is fairly familiar with the field. On about 3 pages, the paper introduces over a 100 symbols, which in itself is already a challenge. I think that it should be possible to somehow reduce the formal burden of the paper. But what is worse, I believe that there are mistakes in the notation. So when the reader arrives at Proposition 1 and the 4 millionth symbols W_i^o and W_i^v, then I think it should be W_i^u instead. In that same proposition, there is a syntactic definition of \delta(W^o, W^u) that has no semantic attached, or at least none that I would understand. Usually explanations follow after the formal definitions, but the intellectual burden for the reader is really enormous, and the authors make little effort to wrap it into didactically well elaborated portions.

**Questions For Authors:**

- if I already had a low-level representation (input of your algorithm), I clearly am in danger to suffer from the potential problem of AIC violation if I map into the abstraction space. Which advantage can I gain from the projection to compensate this risk other than shortcuts (summarizing certain situations in low-level setup)?
- even if I am interested in summarizing variables, couldn't I avoid running into the AIC problem altogether by reasoning on the low level and doing an abstraction only post-hoc?
- in a similar line: If you admit dependency between U-variables, it seems that you start right away from the premise that even the low-level SCM is not described at such a level of detail that noises are independent (as they should be ideally). To me, the projections just seem to make a step into another SCM where there are possibly yet more of those dependencies, and we give it a special name (AIC). The question is: Why is the AIC violation more severe for reasoning than having dependencies among variables in U?

**Relation To Broader Scientific Literature:**

Good.

**Theoretical Claims:**

There are some theoretical statements (even though some or just definitions, like Prop 1). Based on the density of the paper and the lack of clarity, I was not even able to assess what the theorem statements want to say. Sorry, I really made an effort.

Besides, I am not sure about the relevance of the whole question, because the approach starts from the premise to have a low-level SCM already available, which is of course usually not the case. I am partially willing to revise my review (even though I have spent quite a significant time on this paper already to understand what is going on) if the authors answer to my below questions.

---

> ### Author Rebuttal · Authors · 2025-04-01
>
> We appreciate the reviewer's feedback and believe some misunderstandings may have led to a harsh evaluation. We respectfully ask for reconsideration based on the clarifications below.
>
> > In general the paper is very difficult to read [...]
>
> **Response:** We recognize that the reviewer’s primary concern is the paper’s density. While we aimed for clarity through diagrams and examples, some content had to be relegated to Apps. B and C, preventing us from elaborating as thoroughly as we would have liked. The density arises from building upon several works that already involve complex semantic systems, including:
> 1. Pearl Causal Hierarchy (PCH): Our approach is general, encompassing all 3 layers of the PCH. While we chose to present examples using simpler interventional quantities from Layer 2, we present all definitions generally to convey the full scope of our contribution (Layer 3).
> 2. Abstractions: We follow established abstraction theory, using concepts like inter/intravariable clusters, $\tau$-abstractions, AIC, and C-DAGs, with notation aligned to prior literature.
> 3. Causal Generative Modeling: Our work extends neural causal models (NCMs), forming the basis for Sec. 4 experiments.
>
> In the main paper, we prioritized our new contributions over reviewing prior work, referencing established concepts rather than restating them.
>
> That said, we are committed to improving clarity by:
> - adding a notation summary table,
> - expanding discussion on related works,
> - using the extra page to elaborate on each definition.
> We appreciate your willingness to work with us in evaluating the contributions presented.
>
> >I am unsure what significant claims this paper wants to make.
>
> **Response:** We emphasize in our paper that our primary goal is to develop a causal abstraction framework robust to violations of the abstract invariance condition (AIC). When two low-level interventions produce different effects but map to the same high-level intervention, existing frameworks fail to address the resulting ambiguity. This problem is particularly prevalent when working in the representation space after performing representation learning, as a learned, lossy representation is unlikely to avoid such ambiguity. Working in a representation space offers substantial advantages for high-dimensional causal inference, a direction we aim to advance in this paper. We hope this motivation is clearly conveyed in the introduction (lines 58–75, left), supplemented by the example on HDL and LDL cholesterol, as is our proposed solution (discussed from lines 76-95 (left)). We will refine the introduction for clarity and welcome any specific suggestions.
>
> >the approach starts from the premise to have a low-level SCM already available [...]
>
> **Response:** The relaxation of this requirement is the entire contribution of Sec. 3. Recognizing that we are unlikely to have the whole low-level SCM, we introduce identification results (Def. 9, Thm. 3) that specifically aim to infer causal quantities given limited data from the low-level SCM, such as observational data.
>
> > if I already had a low-level representation (input of your algorithm) [...]
>
> **Response:** The benefits of working in the abstraction space are similar to those motivating the use of representation learning in non-causal contexts: reducing dimensionality for improved tractability, transforming data into spaces with desirable mathematical properties (e.g., the linearity of Word2Vec), and better interpretability.
>
> The AIC challenge is unique to the causal setting, as non-causal contexts do not require consideration of how causal relationships between variables are preserved or altered when the representation is lossy. Previously, the trade-off of working in the abstracted space, as you mentioned, meant accepting AIC violation risks for the benefits of representation learning. Our work resolves this dilemma by enabling the advantages of representation learning without incurring any drawbacks from AIC violations.
>
> > even if I am interested in summarizing variables [...]
>
> **Response:** Certainly, but doing so would prevent leveraging the computational benefits of representation learning in the causal modeling process.
>
> >in a similar line: If you admit dependency between U-variables [...]
>
> **Response:** This is a good question. Recognizing that AIC violations can be reinterpreted as projections into the exogenous space is one of the key insights of our paper. Previously, the prevailing view was that if a high-level intervention was ambiguous, performing the abstraction was futile since it meant ignoring details that were relevant to the setting (resulting in mathematical contradictions).
>
> The natural connection to SCM partial projections enabled a generalization of previous frameworks, accommodating AIC violations by interpreting the lost details as exogenous variables. Establishing this formal connection and generalization constitutes a key and non-trivial set of contributions presented in Sec. 2.

---

### Official Review · Reviewer_iNbN · 2025-03-13

**Overall Recommendation:** 4

**Summary:**

This paper introduces a new notion of causal abstraction called "projected abstraction" that extends causal abstraction theory to handle lossy representations—situations where multiple low-level interventions with different effects map to the same high-level intervention. The authors show how to construct projected abstractions from low-level models, translate causal queries between levels, identify high-level causal relationships from limited low-level data, and demonstrate the effectiveness of their approach in high-dimensional image settings. Broadly, this work bridges causal reasoning and abstraction.

**Claims And Evidence:**

The paper claims to address the limitation named the "Abstract Invariance Condition", a common scenario where two variables cannot be abstracted together because they have different downstream impacts. The proposed notion of projected abstraction deals with this issue by representing the loss of information from collapsing two variables that have different downstream impacts in terms of exogenous variables in the high-level causal model.

This is an important direction for causal abstraction theory to be developed in. Exact transformations or abstractions should be very rare, and understanding how to model lossy abstraction will be key in all practical efforts.

**Essential References Not Discussed:**

These aren't essential citations, I believe that the authors did a good job citing the relevant work in causality. However, given that causal abstraction has been applied to mechanistic interpretability, it might be good to situate the paper relative to that literature and explain to readers from that field how this relates.

https://proceedings.mlr.press/v162/hu22b.html

https://proceedings.neurips.cc/paper/2020/hash/92650b2e92217715fe312e6fa7b90d82-Abstract.html

https://arxiv.org/pdf/2502.20914v1

https://arxiv.org/abs/2106.02997

**Experimental Designs Or Analyses:**

The experimental section seems to be good to me, but it will be difficult for readers not familiar with NCMs. I would recommend trying to give the experimental section some more room to breath.

**Methods And Evaluation Criteria:**

The MNIST task in this paper seems appropriate for the proposed methods

**Other Comments Or Suggestions:**

N/A

**Other Strengths And Weaknesses:**

I think introducing the notion of identifiability into causal abstraction is an important aspect of this work, as it helps bridge existing theory on causal abstraction to other research in causality.

Additionally, developing methods for inferring abstract structure from limited concrete low-level data is crucial for causal abstraction to be applied in real world settings.

**Questions For Authors:**

How does this relate to causal feature learning? I don't think you cite any work from that area, but it seems deeply related to me.

**Relation To Broader Scientific Literature:**

This paper is written for theoretical researchers in the field of causality. Within that context, this paper is very strong and will be of great interest. However, I believe that researchers outside this field will find it difficult to work through.

I don't think this is a real issue, and I think ICML should accept theoretical work on causality!

**Theoretical Claims:**

I can confirm that the main text is coherent and the definitions make sense given prior work in the area. No obvious issues stick out. However, I cannot attest to the proofs or appendix material as it would take too much time as a reviewer to go through them all in detail.

---

> ### Author Rebuttal · Authors · 2025-04-01
>
> We thank the reviewer for the positive feedback and review. We are happy that the work was understood. To answer your concerns:
>
> >The experimental section seems to be good to me, but it will be difficult for readers not familiar with NCMs. I would recommend trying to give the experimental section some more room to breath.
>
> **Response:** Thank you for the suggestion. We will expand the experimental section to contextualize NCMs for unfamiliar readers, and we will include a section in the appendix discussing NCMs and their relevant results to the discussion in this paper.
>
> >However, given that causal abstraction has been applied to mechanistic interpretability, it might be good to situate the paper relative to that literature and explain to readers from that field how this relates.
>
> **Response:** Thank you for these citation suggestions; we will be sure to include them. Mechanistic interpretability is a prominent issue in the field of causal abstractions, and we will certainly add an appendix section to discuss it in more detail. Although this work focuses more on the foundational aspects of abstraction theory, particularly performing cross-layer inferences, we find that many of the results are strongly applicable to both problem settings. Relating this work to the mechanistic interpretability problem is a clear next step.
>
> >How does this relate to causal feature learning? I don't think you cite any work from that area, but it seems deeply related to me.
>
> **Response:** Actually, the problem of causal feature learning is highly relevant. Papers such as “Visual Causal Feature Learning” (Chalupka et al., 2014) leverage data-identifiable forms of the AIC to learn ideal representations and variables that best preserve the causal relationships between variables. This connection is briefly discussed in “Neural Causal Abstractions” (Xia & Bareinboim, 2024) Appendix D.2. We will also include a section in our paper complete with these citations.

---

### Official Review · Reviewer_cdvd · 2025-03-17

**Overall Recommendation:** 4

**Summary:**

This paper presents a theory of causal abstractions that generalizes them beyond the usual abstract invariance condition. It formalizes the idea of general causal abstractions through the concepts of partial SCM projections, soft interventions and generalized queries. An algorithm to construct these is general abstractions is obtained. Results regarding inference on the general causal abstractions (through a notion of project cluster causal diagram) are obtained. Some experimental validation of the theory is presented with inference tasks on coloured MNIST datasets.

**Claims And Evidence:**

- The contributions of the paper are mainly theoretical. The theoretical framework is very well supported by useful and insightful results, clear explanations and examples.
- The experimental validation is somewhat convincing although not totally. I do not think that this is a significant weakness of the paper, it is known that the learning problem for causal models is difficult. However, I would like to see a more nuanced discussion of the experimental results (see below). I would also like to see a discussion of the limitation of the experimental setup.
- I would like to see a discussion of the limitations of the proposed theoretical framework, as well as well as future perspectives.

**Essential References Not Discussed:**

Not that I know of.

**Experimental Designs Or Analyses:**

- "The abstractionless model has higher error than the projected C-DAG model since it operates in a higher-dimensional space." I don't find this explanation fully satisfying. As far as I understand, the fact the the baseline operates on a higher-dimensional space could be both beneficial or detrimental. The reduction in dimension is also not dramatic. It is necessary to understand better if this is indeed the cause of the improvement or if there is another explanation.
- In the coloured MNIST experiment one thing is not clear to me, should it even be possible to sample from the right distribution using a binary representation?
- We see an improvement with respect to the non-causal model, but the digits themselves don't look right compared to the 16 dimension RNCM model. I could like to see versions of both models with intermediate numbers of dimensions (for example going from R^16 to R) to see how to performance degrades with less dimensions.

**Methods And Evaluation Criteria:**

The proposed evaluation is appropriate.

**Other Comments Or Suggestions:**

I suggest to nuance/precise this type of statement:
"Combining these two modes of reasoning is vital for building more advanced AI systems."
The word "vital" is strong, so either replace with something weaker or elaborate on what is meant.

**Other Strengths And Weaknesses:**

**Other strengths**
- The work tackles an important problem in an original way
- The paper is very well written and there is a nice effort to make it understandable

**Questions For Authors:**

In definition 7, is there a consistent generalization of the PCH to soft interventions? If yes, would be interesting to discuss/include it.

**Relation To Broader Scientific Literature:**

As far as I know, this paper is a very significant contribution to the literature on causal abstractions. I am not aware of previous works that dealt with relaxing the AIC condition. The results of the paper open the door to applying causal abstractions to much wider problems.

**Theoretical Claims:**

The defintions and statements are clear and sound. I however did not check the proofs in detail.

---

> ### Author Rebuttal · Authors · 2025-04-01
>
> We thank the reviewer for the positive review and valuable insight in our experimental analysis. We address your comments below.
>
> > I would like to see a discussion of the limitations of the proposed theoretical framework, as well as well as future perspectives.
>
> **Response:** Indeed, one of the main limitations of the theoretical framework is that, as a tradeoff for violating the AIC, it is possible that more causal dependencies are introduced, as illustrated through the definition of the projected C-DAG in Def. 8 (more edges are added when there are AIC violators). We will emphasize this point further in the paper. This naturally implies that there is a challenge of determining how to balance the tradeoff between lossy representations and loss of causal constraints, and we will also make a point of this as a potential future direction of work.
>
> > "The abstractionless model has higher error than the projected C-DAG model since it operates in a higher-dimensional space." I don't find this explanation fully satisfying [...]
>
> **Response:** In this particular experiment, the dimensionality reduction is quite extreme. In the $\mathcal{G}$-NCM, the model is trying to sample the entire image $X$ as an intermediate step, while in the $\mathcal{G}_{\mathbb{C}}^{\dagger}$-NCM, the variable $X$ has been abstracted into a binary variable. Since the high-level model has much less information in the input dataset, its only serious advantage is the dimensionality reduction. That said, the error of the $\mathcal{G}$-NCM still trends downward with higher data, since it is theoretically still a sound method of estimating the query. With much more parameters and compute, it is possible that the approach could achieve similar errors. We will include more discussion on this in Appendix D, as well as discussion on the tradeoff of performing such a large dimensionality reduction.
>
> > In the coloured MNIST experiment one thing is not clear to me [...]
>
> **Response:** Indeed, we chose the binary representation as the most egregious example of an AIC violation, and we illustrate one of its clear flaws – one should not be able to achieve any good reconstruction of the image with such a lossy representation. This highlights the strength of our approach of projected sampling – whatever we lose in the representation, we can represent it in the exogenous space. An example analogy is this: suppose we have an image of 10 bits, but we represent it with only 1 bit. We certainly cannot reconstruct a 10-bit image with only 1 bit, but we can reconstruct samples by sampling the other 9 bits. The particular distribution of 9 bits is based on the theory discussed earlier in Sec. 2: we sample conditioned on the 1 available bit and the parents of the variable as if we are translating a high-level intervention to the low-level. We will include this discussion in the paper.
>
> > We see an improvement with respect to the non-causal model, but the digits themselves don't look right compared to the 16 dimension RNCM model [...]
>
> **Response:** Thank you for the suggestion, we will run the two methods at different representation sizes to see how the performance scales. Our expectation is that while the original RNCM degrades in performance as the AIC is more and more violated (translating to worse performance with lower dimensionality), the RNCM with projected sampling does not suffer from this issue. To continue the bit analogy from above, the reconstruction of a 10 bit image would look better if we had 9 of the bits compared to only having 1 bit. However, with projected sampling, we are sampling the remaining bits anyway, so we always have the full 10 bits. Generally speaking, performance of the projected sampling approach could potentially be improved by simply improving the base architecture of the model, although this is out of the scope of our work.
>
> > I suggest to nuance/precise this type of statement: "Combining these two modes of reasoning is vital for building more advanced AI systems." [...]
>
> **Response:** Thank you for the suggestion, we will change it to “Combining these two modes of reasoning unlocks great potential for building more advanced AI systems.”
>
> > In definition 7, is there a consistent generalization of the PCH to soft interventions? [...]
>
> **Response:** Yes, this is indeed an interesting point. We frame Def. 7 in the context where the high-level model is performing hard interventions to emphasize where the noise in the corresponding low-level soft interventions arises as a response to AIC violations. That said, the method could certainly be generalized to handle cases where the high-level model is performing soft interventions as well. Then, the corresponding low-level interventions would be soft interventions that aggregate the results of all of the possible sampled interventions in their high-level counterparts. We will add a discussion on this in the appendix, as you suggested. Thank you!

---

> > ### Comment · Reviewer_cdvd · 2025-04-02
> >
> > I thank the authors for their response. I hope to see an updated experimental section in the post-rebuttal manuscript, as well as a discussion of limitations.
> > Also to clarify the last question, I was also referring to a generalization of the ladders of causality from Definition 2 to the setting of Definition 7.

---

### Official Review · Reviewer_b1c8 · 2025-03-18

**Overall Recommendation:** 3

**Summary:**

This work addresses the problem of causal abstractions, providing an intriguing approach that could extend traditional fine-grained causal applications to more general scenarios. It emphasizes higher-level causal relationships and inferences. A key contribution of this work is the relaxation of the function class limitation in abstraction functions, bridging low-level causal models to high-level ones, and enabling lossy representations known as projected abstractions. Specifically, the work introduces three rules, as outlined in Definition 8 (Partially Projected C-DAG), which play a critical role in handling violations of the abstract invariance condition. Experiments on color MNIST demonstrate the benefits of using projected abstractions.

**Claims And Evidence:**

The proposed method aligns with the claims made in the paper.

**Essential References Not Discussed:**

Related works are thoroughly discussed in this paper.

**Experimental Designs Or Analyses:**

The provided analyses are generally well-conducted.

**Methods And Evaluation Criteria:**

Empirical evaluations on toy data validate the benefits of the proposed method. Here, in the case of color MNIST, which is relatively simple and serves as toy data, the significant advantages of the proposed method are not fully demonstrated. While the importance of relaxing the abstraction function is clear, especially for real-world applications, the simplicity of the dataset limits the verification of these benefits.

**Other Comments Or Suggestions:**

none

**Other Strengths And Weaknesses:**

Strengths:

The organization is well-structured, and the presentation is relatively clear.

It addresses a very important theoretical problem, and the problem is also significant in practice.

I really like the three rules in Definition 8, as they play an important role in 'correctly' constructing high-level causal models.

Some examples are provided, which greatly aid in understanding from an intuitional viewpoint.



Problems:

1) The Abstraction Function, as defined in Definition 4, is required to satisfy certain assumptions to maintain specific structures for inter/intravariable clusterings. This implies that the function class of the Abstraction Function is constrained within an equivalent class. While this is clear from the definition, I am concerned about how such a function class is restricted.

2) The three rules in Definition 8 are key factors in the proposed method. I am curious about the underlying motivation for introducing these three rules. What is the reasoning behind their inclusion, and how do they contribute to the overall approach?

3) Overall, this work mainly addresses AIC violations, but I am concerned whether the results hold regardless of the degree of AIC violations. Does the method remain effective when AIC violations are more significant, or is its performance sensitive to the extent of these violations?

4) My final concern is regarding the experiment settings. While I understand that this work is primarily theoretical, the motivation for addressing lossy representations stems from a practical viewpoint. In this context, the experiments are conducted at a relatively simple level, which raises concerns about the effectiveness of the proposed method in more complex scenarios.



Suggestions:

1) While I can understand the overall story, I feel that the writing style may not be very accessible for those who are not familiar with this area. It would be helpful to provide some intuitive interpretations for the basic and key definitions. For example, a brief discussion of Definitions 3 and 4 from an intuitive standpoint could enhance clarity.

2) It would be better to introduce, from an intuitive (high-level) standpoint, why consistency estimation is still possible when faced with lossy representations, and how the proposed method addresses this issue.

**Questions For Authors:**

none

**Relation To Broader Scientific Literature:**

The paper tries to work on an important problem.

**Theoretical Claims:**

I reviewed the theoretical aspects at a high level but did not rigorously verify the correctness of the theorems.

---

> ### Author Rebuttal · Authors · 2025-04-01
>
> We thank the reviewer for the positive review and detailed suggestions. Addressing the reviewers points:
>
> > The Abstraction Function, as defined in Definition 4, is required to satisfy certain assumptions [...]
>
> **Response:** Yes, broadly, the paper is focused on a specific family of abstractions called constructive abstractions. Most works found in this literature, including ours, focus on this family because there is a natural mapping between low- and high-level interventions and distributions, allowing for well-defined inferences. These abstractions are most easily realizable in practice as well, since the construction of the high-level model naturally follows from the declaration of the clusters, which are interpretable and can also sometimes be learned (see “Neural Causal Abstractions” (Xia & Bareinboim, 2024).) If one would prefer to allow unrestricted abstraction functions, the results of the paper may still be applicable, but one would have to define how low-level interventions correspond to high-level interventions on a case-by-case basis (i.e., it is not necessarily the case that an intervention on $do(\mathbf{X}_L = \mathbf{x}_L))$ on the low-level is equivalent to an intervention on $do(\tau(\mathbf{x}_L))$ on the high-level, or even well-defined).
>
> > The three rules in Definition 8 are key factors in the proposed method. I am curious about the underlying motivation [...]
>
> **Response:** Indeed, that’s a cool question, thank you. The three rules demonstrate the core insight of allowing lossy abstractions – by losing some information in the variables, we must make up for it by removing causal constraints (i.e., adding more edges). These rules are necessary for guaranteeing the validity of the downstream causal inferences, as shown through Thm. 2. Without adding these edges, one risks claiming that a causal query is identifiable when it is not. For example, in Fig. 1(a), if the edge from $Z$ to $Y$ is not included, one may mistakenly assume that $P(Y \mid do(X), Z) = P(Y \mid do(X))$, which is the mistake made in Ex. 2. The proof of Thm. 2 shows the math behind what could go wrong when each of these rules are ignored.
>
> > Overall, this work mainly addresses AIC violations, but I am concerned whether the results hold regardless of the degree of AIC violations [...]
>
> **Response:** This is a great question. In fact, this is precisely the issue that we are solving. In the past, approaches that rely on the AIC would deteriorate in performance as the AIC becomes more and more violated. In contrast, the projected abstraction approach does not care how much the AIC is violated and performs identically regardless. Note, for example, in Fig. 4 of our MNIST experiment, reducing the representation size to a mere binary variable strongly violates the AIC and therefore results in very poor performance from the RNCM (3rd row), but with projected sampling, we can produce images as usual (4th row).
>
> > My final concern is regarding the experiment settings. [...]
>
> **Response:** Our goal with the MNIST experiment was to provide a proof-of-concept of how one could potentially learn a causal generative model without worrying about AIC violations. More generally, this would allow for a unification of out-of-the-box representation learning methods and causal generative modeling methods since it would not matter how the representation is learned once we move to the causal generative modeling phase. With this unification understood, scaling to more complex datasets is a matter of scaling the architecture and representation learning methods using state-of-the-art techniques from general deep learning research, which can be done separately from the causal results guaranteed by our work.
>
>
> > While I can understand the overall story, I feel that the writing style may not be very accessible [...]
>
> **Response:** We aimed for a more rigorous analysis of this highly technical topic for the sake of concreteness, so its more mathematical nature is somewhat to be expected. We do appreciate the suggestion and will add some clarifying sentences in the final report and include an appendix section discussing some of the key prior works to help bring readers up to speed with the contents of the paper.
>
> > It would be better to introduce, from an intuitive (high-level) standpoint, why consistency estimation is still possible when faced with lossy representations, and how the proposed method addresses this issue.
>
> **Response:** Yes, thank you for the suggestion. We will include the discussion mentioned above on the tradeoff between the lossiness in the abstracted variables and the loss of constraints in the projected C-DAG and why inferences in the new model still apply.

---

> > ### Comment · Reviewer_b1c8 · 2025-04-04
> >
> > Thank you for your response.
> >
> > Some points have been addressed in the rebuttal—I appreciate your effort. However, I also reviewed the comments from other reviewers, particularly Reviewer iNbN. To some extent, my concerns regarding the writing style are shared by Reviewer iNbN. For now, I maintain my current rating. I will participate in the next stage of discussion with the AC and the other reviewers, particularly with regard to the writing style.

---

### Official Review · Reviewer_8K54 · 2025-03-25

**Overall Recommendation:** 4

**Summary:**

Existing causal abstraction frameworks often struggle with lossy abstraction functions, where different low-level interventions produce distinct effects but map to the same high-level intervention. To address this, the authors propose projected abstractions, a new framework that extends previous definitions to handle lossy representations more effectively. The construction of projected abstraction and inference is shown.

**Claims And Evidence:**

The claim that the Abstract Invariance Condition (AIC) can be easily violated and invalidate causal conclusion is well supported by the example and explanations. The discussion on when and how lossy transformation should be used is interesting. This highlights the importance of the problem that the paper is tackling.

**Essential References Not Discussed:**

N/A

**Experimental Designs Or Analyses:**

I checked the two experiments and they look correct to me.

**Methods And Evaluation Criteria:**

The abstraction projection looks like a practical method to address the problem of AIC violation.

**Other Comments Or Suggestions:**

N/A

**Other Strengths And Weaknesses:**

The paper is notational heavy and contains a number of definitions that require the readers to consume, especially in Sec 1.1 and 2.1. They are not the easiest to follow. It would be helpful for the authors to state briefly why those definitions are needed and how they will be used. Aside, it is hard to cross-reference the notations from all different parts in the paper to Alg 1. I suggest the authors add explanations to it.

**Questions For Authors:**

1. The details of the theoretical contribution are unclear. It seems like projection has been proposed in Lee & Bareinboim, 2019, but partial SCM projection seems novel in this paper. However, is definition 6 original or discussed in prior work? Could the authors articulate what is novel in the paper in comparison Lee & Bareinboim, 2019? Since the paper’s main contribution is a formalism framework that generalizes from previous literature, at a higher level, I think it would be better to have a separate paragraph to summarize the theoretical contribution.

2. For experiments, what does the variance in each method (Fig 5) come from? It seems like the projected C-DAG approach has a larger variance than other methods. Is it some sort of bias-variance trade off and/or can authors discuss it?

3. What is the scalability of the proposed method in a more complicated, real-world scenario? MNIST is a classic but relatively simple dataset.

**Relation To Broader Scientific Literature:**

The key contribution seems to be a generalization from the prior literature. I think it is a useful addition to the direction of causal representation learning.

**Theoretical Claims:**

I have checked until Def 6.


L110-118 could be put into Def 2, as I got confused by undefined notations when reading Def 2.


Def 5: $\tilde{pa}_V$ is not used.


Def 6: Definition of $\tau$-abstraction is missing in the main text (please move or point to Def 10 in Appendix A).

---

> ### Author Rebuttal · Authors · 2025-04-01
>
> We thank the reviewer for the insightful questions and positive review. We appreciate the multiple suggestions and will address them as follows:
>
> > L110-118 could be put into Def 2, as I got confused by undefined notations when reading Def 2.
>
> **Response:** We will adjust Def. 2 to accommodate this and make it clearer; thanks!
>
> > Def 5: $\widetilde{\mathbf{pa}}_V$ is not used.
>
> **Response:** Thanks for pointing this out. It was originally used in settings in which the AIC was not violated, but we do not consider such cases in the main body, so we will omit this from the definition to improve legibility.
>
> >Def 6: Definition of $\tau$-abstraction is missing in the main text (please move or point to Def 10 in Appendix A).
>
> **Response:** We will include this as suggested.
>
> >The paper is notational heavy [...]
>
> **Response:** Thank you for the feedback. We do assume some background of the previous literature, but indeed, our efforts to fit all of our content into the page requirement have unfortunately led to the paper being a bit dense. To remedy this problem, we have included Appendix B for further discussion and Appendix C for clarifying examples. If accepted, we will use the extra page to clarify each definition and make the theory easier to follow. We will also add a table in the appendix to clarify all of the notation we use throughout the paper.
>
> Finally, to answer your questions:
>
> > 1. The details of the theoretical contribution are unclear. It seems like projection has been proposed in Lee & Bareinboim, 2019 [...]
>
> **Answer:** Yes, SCM projections were proposed in Lee & Bareinboim (2019). We draw inspiration from this result to generalize to partial SCM projections (one of our contributions). In contrast to Lee & Bareinboim (2019), which solves a causal RL problem to find optimal intervention sets in a causal bandits setting, we work in an entirely different space, applying projections to define a causal abstraction formalism. Def. 6 and Thm. 1 (both contributions of our paper) establish this intriguing duality between partial SCM projections and causal abstractions, which is surprising and may otherwise seem unrelated. The idea that a lossy abstraction that violates the AIC can be thought of as partially projecting away the abstracted variables is the main insight that allows us to generalize previous abstraction inference results to new settings where the AIC does not necessarily have to hold. This insight leads to further results about how to perform more general causal inferences across abstractions (Sec. 3) and a new type of sampling procedure for high-dimensional causal generative modeling (Sec. 4). In short, the generalization of SCM projections to the partial form is indeed a contribution of the paper, but the main motivation for this contribution is to generalize challenging instances of causal abstraction inference. We thank the reviewer for pointing this out, and we will include this discussion in the paper's appendix.
>
> > 2. For experiments, what does the variance in each method (Fig 5) come from? [...]
>
> **Answer:** The variance for each approach arises from the sampling of the training data and the randomness in the training process (e.g., parameter initialization). One explanation for the seemingly higher variance in the projected C-DAG approach may simply be that the plot somewhat exaggerates it due to the log-log formatting. Another deeper reason may be that there is a tradeoff between granularity and constraints when we convert from the original C-DAG to the projected C-DAG (as we are potentially adding more edges). With more edges, there are fewer constraints; therefore, converging may be more challenging despite the higher accuracy. Of course, this more nuanced view is just brought up by the new machinery developed in this paper. This paper is the first to study relaxed abstractions under violations of AIC, which are very likely to occur in almost any scenario.
>
> > 3. What is the scalability of the proposed method in a more complicated, real-world scenario? [...]
>
> **Answer:** One issue that is resolved by our paper is that the AIC can be a challenging thorn when attempting to apply out-of-the-box representation learning tools in causal generative modeling contexts. We would like to be able to learn an NCM on top of any representation, but the AIC prevents us from doing so. The goal of the MNIST experiment is to show a proof of concept that, regardless of how the representation arises, we can still perform causal inferences and sample causal images using the new projected-sampling approach. This is powerful as it means that the method can achieve robust performance regardless of the method of representation learning used. Scaling to even higher-dimensional settings may require larger architectures, but fortunately, one can leverage the vast research of the general deep learning community to accomplish this while still enjoying the causal benefits from this work.

---

> > ### Comment · Reviewer_8K54 · 2025-04-01
> >
> > Thanks for the response. It addresses my concerns. I particularly like the point ``duality between partial SCM projections and causal abstractions'', so I'm increasing my score from 3 to 4.

---

### Decision · Program_Chairs · 2025-05-01

**Decision:**

Accept (poster)

**Comment:**

This paper proposes a novel framework for causal abstraction under lossy representations by introducing projected abstractions, generalizing causal abstraction models to handle violations of the Abstract Invariance Condition (AIC).

The paper was well received by most reviewers for its originality, theoretical depth, and connection with practical causal inference challenges. While the empirical evaluation was somewhat limited in scope, it served as a convincing proof of concept. A few reviewers noted that the notation and density of definitions made the paper challenging to follow, and suggested improvements in clarity, intuitive explanation, and experimental elaboration. Reviewer jmSk provided the only negative review, due to the presentation issues.

Despite concerns about the presentation raised by multiple reviewers, and one reviewer providing a negative score primarily due to clarity,  the majority of reviewers agreed that the paper addresses an important gap in causal abstraction and offers meaningful theoretical and practical advances while finding the ideas well-motivated and clearly articulated.

Given the significance of the problem, the originality of the proposed solution, and the overall positive reception, the AC recommends accepting this paper. The authors should incorporate the reviewers’ constructive suggestions on the presentation to improve readability in the final version.